# From Brain Lobes to Neurons: Navigating the Brain Using Advanced 3D Modeling and Visualization Tools

**DOI:** 10.3390/jimaging11090298

**Published:** 2025-09-01

**Authors:** Mohamed Rowaizak, Ahmad Farhat, Reem Khalil

**Affiliations:** 1Department of Architecture, Art, and Design, American University of Sharjah, Sharjah P.O. Box 26666, United Arab Emirates; mohamed.rowaizak@gmail.com; 2Dioscuri Centre in Topological Data Analysis, Mathematical Institute, Polish Academy of Sciences, 00-901 Warsaw, Poland; ahmad.farhat.research@gmail.com; 3Department of Biology, Chemistry and Environmental Sciences, American University of Sharjah, Sharjah P.O. Box 26666, United Arab Emirates

**Keywords:** educational neuroscience, 3D modeling, visualization, human brain, technology, software, instructional strategies

## Abstract

Neuroscience education must convey 3D structure with clarity and accuracy. Traditional 2D renderings are limited as they lose depth information and hinder spatial understanding. High-resolution resources now exist, yet many are difficult to use in the class. Therefore, we developed an educational brain video that moves from gross to microanatomy using MRI-based models and the published literature. The pipeline used Fiji for preprocessing, MeshLab for mesh cleanup, Rhino 6 for target fixes, Houdini FX for materials, lighting, and renders, and Cinema4D for final refinement of the video. We had our brain models validated by two neuroscientists for educational fidelity. We tested the video in a class with 96 undergraduates randomized to video and lecture or lecture only. Students completed the same pretest and posttest questions. Student feedback revealed that comprehension and motivation to learn increased significantly in the group that watched the video, suggesting its potential as a useful supplement to traditional lectures. A short, well-produced 3D video can supplement lectures and improve learning in this setting. We share software versions and key parameters to support reuse.

## 1. Introduction

Neuroscience research has enhanced our understanding of the brain’s structural and functional complexities that give rise to cognition and behavior [1,2,3]. Neuroscience research has the potential to drive medical advancements [4,5], improve quality of life [6,7], and address major questions regarding the nature of the mind [8,9,10]. As such, neuroscience education is an important field that merits attention. Moreover, neuroscience is integral in educational settings [11,12], offering insights into teaching and learning strategies. Consequently, this enables educators to understand how the brain learns and develops, resulting in improved instructional techniques [13]. 

Neuroscientific instruction has relied on 2D static diagrams, which show one view and lose depth. High-resolution 3D resources now exist, including the Human Connectome Project (HCP) [14], BigBrain Project and Allen Brain Atlas [1] and Visible Human Project [15]. These enable students to interact with the brain by rotating, scaling, and panning, but are difficult to navigate and use. Therefore, we created an engaging 3D video of the brain with enhanced elements of dimension, textures, lighting, materials, and effects as a supplement to classroom lectures. We introduce several advanced 3D modeling and visualization tools, such as Rhino 6 [16], Houdini [17,18], Cinema4D [19], and After Effects [20], that are used in commercial and professional production studios. Where possible, we favored peer-reviewed, widely adopted open-source tools to improve transparency and reuse. We used Fiji (Version 8) [21] as a bridge between MRI scans and digital brain structures, and MeshLab (Version 2022) [22,23], Rhino 6, Cinema4D, and Houdini to model, sculpt, simulate, and verify digital objects. Finally, After Effects was used to combine the output of all software for editing and visual consistency to create a video. This innovative approach combines multiple techniques to create high-resolution models as neuroscientific visuals.

We chose this pipeline, using these specific tools, for several reasons. The software Rhino builds accurate 3D shapes with NURBS (Non-Uniform Rational B-Splines), so dimensions match real measurements. It also reads and writes common CAD and mesh formats like STEP, IGES, STL, OBJ, and DWG, which keeps it easy to move models between tools. For fabrication, Rhino has the advantage of checking tolerances and creates watertight meshes, so models pass 3D-printing and CNC checks without extra cleanup. The software Houdini uses a procedural node graph that records every step, which makes edits safe and repeatable even late in a project. It runs high-quality simulations for smoke, fluids, cloth, and rigid bodies, with strong performance on modern GPUs and CPUs. Lastly, Cinema 4D has a clear interface and predictable tools that new users can learn quickly. The procedural system in Cinema 4D, MoGraph, allows the user to clone, animate, and vary thousands of objects using simple effectors, which is ideal for motion graphics. It connects directly to Redshift for fast GPU rendering and links well with After Effects for compositing. It also offers granular control over neuron sculpting, distinguishing it from other software like Zbrush, which, while adept at sculpting, lacks animation features.

The final video has annotations and visual effects to facilitate transitions between labels and content. The initial emergence of the brain was used as a tool to visually provoke the viewer and spark curiosity. We used advanced video editing software to overlay text on frames, enhancing the clarity of the content. This method offers flexibility in content creation and production. By generating these 3D models, we pave the way for integration with emerging educational technologies such as immersive Virtual Reality, aligning with both current and future educational trends. We tested whether adding a short 3D brain video to a standard lecture improves student comprehension and motivation relative to lecture alone. We predicted a larger pre–post gain in test scores and higher posttest motivation ratings in the video group.

## 2. Materials and Methods

Here we provide a detailed description of the complex process used to digitally produce the video (Figure 1). We also report software and parameter settings in Appendix A. MRI data from a healthy adult were acquired from an open-source platform called IMAIOS [24]. We used these scans to build the gross brain 3D model. The pipeline also works with other public acquire MRI datasets and with Freesurfer processed volumes [25]. We utilized Fiji, an open-source application that offers a wide range of libraries and plugins specifically relevant to biological research [26]. In our study, Fiji was primarily used for stacking MRI scans to create a 3D model based on the 2D scans. This 3D model is then exported as a mesh model, composed of vertices, edges, and faces, combined to form scalable 3D objects.

Next, we imported the mesh model into MeshLab, a mesh processing software [22]. MeshLab was employed to edit, clean, and inspect the imported model, ensuring its accuracy and quality. MeshLab was used to repair and decimate meshes with batch filters that keep steps reproducible [23], while in other software like Blender, the cleanup is manual and less transparent for logs. Once this step was completed, we proceeded to Rhino 6, a well-known 3D design software. In Rhino 6, we utilized its rendering capabilities to enhance the visual appearance of the model and embed materials into it. Rhino 6 with Grasshopper provided precise NURBS edits when manual fixes were needed, though it requires a license and can feel heavy on very polygon-dense models. The file format exported from Rhino (.obj) can store basic material characteristics and textural mapping on the mesh. The actual material assigned was accomplished in Houdini, along with the camera and lighting setting, to curate the visual experience for the viewer. To generate other components of the brain that required more granular control of variables, we chose SideFX Houdini 18.5.596, a software with procedural modeling features. This allowed us to manipulate and create these components with precision and flexibility. Houdini provides a node-based graph that records every parameter, which makes the modeling, shading, and lighting path fully specified and reproducible. While geometry nodes in Blender can approximate this, our team already maintained a stable Houdini graph for this project and achieved consistent renders.

Simultaneously, we also employed Cinema 4D 2023 for custom modeling and detailing of the neuron and the cortical column [19]. Cinema 4D is a user-friendly animation platform that offers a unified platform for modeling and rendering. We were able to generate an accurate model of the neuron’s morphology by referencing the NeuroMorpho.org repository [27]. Finally, in Adobe After Effects (Version 2022), we compiled all the rendered frames and animations to create the video [20]. We incorporated annotations, labeling, and visual effects to enhance the final video. Given these trade-offs, Fiji handled image preprocessing, MeshLab handled cleaning, Rhino handled targeted manual corrections, and Houdini handled the procedural scene and final renders. This pipeline minimized risk, maintained traceability, and met the visual quality needed for teaching.

The following is the sequence of steps that were implemented to generate the video:Step 1: Downloaded raw images from the source.Step 2: Edit the raw images in Photoshop to isolate the brain from remaining structures of the head.Step 3: Import the edited image sequence into Fiji.Step 4: Stack the images in Fiji to create a 3D form.Step 5: Export the 3D form from Fiji to a .obj file format.Step 6: Import the 3D form into Rhino 6 for visual inspection.Step 7: Import the 3D form into Meshlab to apply post-processing, which includes cleaning the model, smoothing the surface, and then exporting again.Step 8: Import the 3D form into Houdini Fx.Step 9: Apply and animate an effect to slice the 3D form.Step 10: Export the animation as frames in .png format.Step 11: Apply and animate the painting and rotation of the 3D form.Step 12: Export the animation as frames .png format.Step 13: Apply and animate the growth form effect into the 3D form.Step 14: Export the animation as frames in .png format.Step 15: Apply neuron sculpting and animation in Cinema 4D.Step 16: Export animation as frames in .png format.Step 17: Create cortical column assembly in Cinema 4D.Step 18: Export animation as frames in .png format.Step 19: Import all rendered frames into Adobe After Effects.Step 20: Compile the imported frames in a layered sequence.Step 21: Apply annotation, color grading, and animations imported files.Step 22: Export as a single edited video in .mp4 file format.

### 2.1. Data Collection

The specific MRI scans retrieved from IMAIOS [24] were obtained from a horizontal and coronal series of a healthy individual. The resolution of the brain increases with the number of stacked images. We used over 130 MRI scans stacked at intervals of 0.6mm, with a reasonable resolution, to generate a three-dimensional shape illustrating the main features of a healthy brain. This stacking process was carried out using Fiji Image J, an open-source image processing platform capable of transforming a stack of images in .png format into a 3D object [26]. Figure 2 displays sample MRI scans of the brain in the horizontal plane, with anatomical structures revealed at varying depths.

In addition to MRI, we used atlas figures and Neuromorpho exemplars to model neurons. We videoed the gross anatomy and fine features such as cortical columns and neuron morphology. We referred to [28,29] to delineate the different layers of gray matter in a cortical column. MRI scans were converted into 3D objects using Fiji, which utilized white, gray, and black pixels to represent solid and void areas, respectively. Adobe Photoshop (Version 2022) [30], which recently incorporated AI technology, was used to isolate the brain from other surrounding elements, allowing detection of edges with better precision. Figure 3A illustrates the use of the ‘object selection tool’ to select specific components of the image and then refine the selection with a manual ‘eraser’. The dashed white square in Figure 3A is shown at a higher magnification in Figure 3B to emphasize the regions selected with the object selection tool. Importantly, Figure 3C,D depict the raw downloaded file from IMAIOS and the edited version, respectively, revealing a noticeable difference between the images.

The edited images were then stacked in sequential order in Fiji, to ensure that they are imported and arranged correctly. The images are processed to construct a 3D model by initially importing them as an ‘image sequence’, adhering to the order established during saving. The ‘Image Stack’ feature of Fiji was used to transform these images into a 3D structure, which then automatically processes and exports them in the preferred file format. The resulting 3D model is directly derived from pixels to voxels, presented as a rough structure with unpolished edges and faces. The model’s resolution is determined by adjusting multiple parameters in the settings and defining the extrusion thickness for each image based on its light-colored pixels. The final model is then exported in the .obj file format, which inherently captures the mesh attributes embedded within the model.

### 2.2. Computer-Aided Modeling

Creating and modeling brain structures at varying scales required the use of different software programs, each tailored for specific detailing, sculpting, and manipulation tasks. The model that was exported from Fiji as a .obj file served as a raw mesh, which required further refinement. This mesh was first reviewed in Rhino 6, where its quality was assessed to determine subsequent actions in MeshLab [16,22,23]. The model was analyzed for disjoint meshes, missing faces, irregular face normals, and other characteristics that create an inaccurate mesh, making it difficult to work in subsequent steps of the process. Some of the issues identified in the model were fixed in Rhino and others were fixed in Meshlab. A useful feature in Rhino is the ‘delete disjoint mesh’, which automatically eliminates extraneous 3D fragments not integral to the primary brain model. These fragments often arise from inaccuracies in earlier image processing or residual unconnected white pixels. Figure 4 shows the interface and application utilities that we used to generate the brain model in Rhino 6.

The analysis of the mesh, along with visual observations, revealed an uneven surface and missing faces that resulted in voids and discoloration across the model. Therefore, the exported 3D model from Rhino 6 was then refined using MeshLab. We used a range of editing techniques to prepare the model for animation and rendering, including key functions such as cleaning and repairing, repairing non-manifold geometry, removing isolated pieces, applying remeshing and simplification filters, uniform mesh resampling, cleaning vertices, surface reconstruction, decimation, and smoothing. The main functions we used were ‘Remove Isolated’, Simplification: Quadric Edge Collapse Decimation, and various mesh cleaning and reconstruction filters.

For post-processing, the model was re-imported into Rhino 6 to ensure that the modifications aligned with our intended representation of specific brain regions [16]. Rhino 6’s user-friendly interface and clear graphics made it an ideal choice for this verification step, primarily focusing on the brain’s shape to ensure that we did not miss any inaccuracies or defects in the model. We judged completeness of the model by visual match between MRI scans and the 3D brain, with clear delineation between anatomical structures.

For finer modeling and sculpting of brain structures, Cinema4D was used for this step [19]. This software is widely known for its capabilities in character design and environmental effects, as it replicates realistic lighting, materials, textures, and movement. We referenced 3D reconstructions of neurons that are publicly available on the NeuroMorpho.org repository [27] to aid in the sculpting process of the pyramidal cells and accurately depict its morphology. Our neuron sculpting process in Cinema4D began with basic platonic cylindrical shapes assembled to capture the neuron’s high-level form. This form is characterized by a long spine with a pronounced diameter on one end and branching structures that decrease in size and thickness as they extend from the central spine.

### 2.3. Procedural Modeling

To create the 3D model of the brain, we used procedural modeling in Houdini. This is an advanced technique used to create 3D models and textures from sets of rules. Houdini works by strategically devising techniques that result in a set of rules that feed each other data to simulate a process, scenes, or elements and ultimately creating an exportable 3D model. Understanding the basic knowledge on brain growth, then oversimplifying the process into a gamified version for an engaging visual, entails high-level planning of the file, as well as preparing the 3D models, lighting, and camera. These animations and visualizations can be rendered as high-definition frames and subsequently compiled into a video. Below is a detailed breakdown of the components used to setup the file:Geometry: Within this component, we integrated models from prior steps and executed specific actions. For instance, ‘slicing’ was employed to segment the model, revealing particular sections, while ‘paint’ facilitated manual selection and animation of brain regions.Light: We chose specific lighting types to accentuate the model’s textures and details. Both ‘Area Light’ and ‘Spot Light’ were utilized, with adjustments made to their color, intensity, and spatial positioning.Camera: This represents the rendering scene’s viewpoint. We fine-tuned various parameters, including aperture, focal length, and clipping plane, to achieve the desired rendering effects.Material Palette: Within the ‘geometry’ component, the selected object’s material can be tailored. Our material strategy aimed to replicate a realistic appearance.

### 2.4. Video Compilation and Editing

Finally, we used Adobe After Effects to compile and edit the video by importing the exported frames and arranging them sequentially. During the editing step of the video, we added annotations to define different anatomical structures as well as apply visual effects and animation to make the video more engaging. The image frames and videos serve as the result of previous steps, remaining in their raw format without any editing, effects, annotations, labeling, or comments. In the final video, there were obvious variations in color tones, which was due to the use of different software. It is important to maintain a consistent file format by setting the properties of the composition, which includes determining the resolution and duration of the final video export, thus ensuring that the exported file maintains the desired resolution and quality after editing. Figure 5A provides a screenshot illustrating the selected specifications for the composition export and the resulting video quality. After finalizing the video editing phase, a standard video file was subsequently generated, which is compatible with most viewers. Setting the width and height of the composition allows all imported media to maintain the same resolution and quality. The aspect ratio of 1280 × 720 is a standard media industry resolution that aligns with most monitors and projectors used in classrooms. Similarly, the frame rate selected for 30 frames per second is higher than the standard of 24 frames per second, which allows for more control over transitions between imported media and applying animations across the media content. In the next section, the start timecode and duration specify a timeline of 45 s for the length of the exported video. Finally, the background color helps homogenize the void space in the content uploaded.

After importing the site files into Adobe After Effects, they are organized in the ‘Project’ window on the left side of the screen. Subsequently, they are dragged and dropped into the lower window, arranged sequentially for easy access during editing. Figure 5B shows the imported content in the top left window and the arranged composition in the bottom window. The layers show a timestamp in the timeline section, with their imported duration, if imported as video or frames, or with a placeholder that can be stretched to the required duration and shifted to be placed at the desired time. Under each of the layers, a dropdown menu can be clicked to further edit the media with effects, animations, and specific key-frame controls. The information section on the right side of the screen is the properties panel for customizing characteristics and attributes of the layer selected; an example is changing the font, color, and size of text.

Annotations and titles were animated to create engaging graphics. Once the composition is complete, the edited film was then exported in the desired file format for viewing on any device. The colored bars in the timeline window represent the original duration of each content, which was adjusted to align with the overall flow of the video. Additional edits, such as the rotation of the brain, were applied to specific content to enhance the visual experience.

### 2.5. Study Design and Hypotheses

We ran a single-session two-arm randomized study in a human biology course for non-majors. Students gave informed consent at the start. The Institutional Review Board at the American University of Sharjah approved the study (approval #24-019). We enrolled 96 students who met inclusion criteria. We excluded only those who did not provide consent or were younger than 18. After obtaining consent, we randomly selected 46 students in the classroom to form the control group and asked them to move to an adjacent room. The remaining 50 students formed the experimental group. There was one session with no follow-ups. See Appendix A for the CONSORT diagram.

Both groups received identical content coverage and equal total instructional time. The control group received the standard lecture on brain structure and function. The experimental group watched a brief 3D video integrated with a lecture, adjusted so total time matched the control condition. Content matched across arms. This controlled for time and content, so any difference reflects delivery mode. We assessed comprehension and motivation with a brain knowledge survey administered in Google Forms. The survey contained ten multiple-choice items of varied difficulty that map to the lecture and video learning objectives. It also contained four Likert items on motivation and perceived usefulness. Students completed the same survey at the start, pretest, and at the end, posttest. We scored tests from anonymized IDs.

The primary outcome was the pre-to-post change in total test score on a 0 to 100 scale. Secondary outcomes were item accuracy and the four Likert measures of motivation and usefulness. The experimental group exceeded controls in posttest scores on gain, motivation, and usefulness. Data were analyzed to assess changes in Likert scale ratings pre and post intervention for both control and experimental groups. Ratings were treated as ordinal outcomes, and two complementary statistical approaches were applied. An ordinal generalized estimating equations (GEE) model with a cumulative logit link was fitted to estimate the odds of achieving higher Likert scores. The model included group (experimental vs. control), time (posttest vs. pre-test), and their interaction, with Likert items as a covariate to account for differences in item response tendencies. Odds ratios (ORs) and 95% confidence intervals (CsI) were derived from the model coefficients to quantify the magnitude and direction of effects. This approach was selected to account for the correlated nature of repeated measures within participants and to provide population-level estimates across all Likert items.

In addition to the GEE, within-group changes from pre to post test were examined using Wilcoxon signed-rank tests for each Likert item. This non-parametric method was chosen because it does not assume normality of score distributions and is robust to the ordinal nature of the data. For each participant, the difference score (Δ = post–pre) was calculated, and the median change was tested against zero. This approach allowed for direct assessment of improvement patterns within each group, providing an item-by-item complement to the pooled GEE results.

### 2.6. Anatomical Validation

This video is an educational resource for undergraduate biology, psychology, and neuroscience courses, and is not intended for diagnosis, surgical planning, or other clinical use. Two neuroscientists conducted independent, side-by-side reviews of our meshes against Allen Brain Atlas and NeuroMorpho exemplars. For gross anatomy, they compared lobe boundaries and major sulci to atlas figures [1]. For neurons, they compared soma size, primary branch counts, and overall arbor architecture to NeuroMorpho exemplars [27]. Reviews were performed separately, after which we discussed any differences. We reached agreement that the models were anatomically plausible for teaching and matched the intent of the video. This approach is qualitative and literature-based; it does not provide quantitative accuracy estimates and should not be used to support clinical claims. Furthermore, future work for clinical use would require patient-specific high-resolution data, manual quality checks, and quantitative validation for atlas overlap and boundary error.

## 3. Results

### 3.1. Gross Brain Structure

The video link is found in Appendix A and includes a general overview of the gross brain anatomy with annotations for the four lobes of the brain and their corresponding function (Figure 6A). The model is then rotated 360 degrees in the video, allowing the viewer to observe the full extent of the gross anatomy. To ensure the quality and accuracy of the brain model, the Human Brain .obj file (Appendix A) is finally imported into Rhino 6. In Rhino 6, the model is visualized in a 3D form, allowing rotation, zooming, and slicing to visualize and inspect the quality of the final exported model. The video frame shown in Figure 6A demonstrates this. In this frame of the video, the brain lobes are presented in different colors, and annotations of each lobe with their corresponding functions are presented. This figure also features the next part of the video, which reveals the gray and white matter of the brain through successive sections in the coronal plane (Figure 6B).

### 3.2. Cortical Column

The next part of the video presents the viewer with a cortical column and its corresponding annotation reflecting its function. The cortical column is the basic functional unit of the cerebral cortex, and columnar organization is thought to be important for information processing. The seminal work of Hubel and Wiesel [31] and Mountcastle [32] showed that neurons vertically organized through the layers of the cortex (from the white matter to the surface of the brain) often share similar functional properties. Since the cortex comprises six layers, each with distinct types of neurons and functions, we modeled the cortical column, illustrating the unique laminar organization of cells. We show how a cortical column typically encompasses neurons in all these layers. The size and type of cells were referenced from Thompson [33] to guide the sculpting. Figure 7A illustrates the assembly of five neuron models to represent the cortical column across the six layers.

It is important to note that the assembly of these models into column structures serves as a simplified representation of the actual density of neurons in a column in the human brain. To aid visual comprehension, different colors are assigned to each neuron, facilitating differentiation and identification. Figure 7A shows pyramidal neurons in layers 2, 3, 5 and 6, and a stellate cell is found in layer 4. Once the cell models have been created and manipulated, they are carefully positioned in specific locations across the layers to depict the concept of a cortical column. Figure 7B illustrates a representative cortical column with laminar boundaries and corresponding cells in each layer.

### 3.3. Pyramidal Neuron Morphology

In the next part of the video, we sought to highlight the major neuron type in the cerebral cortex that gives rise to interareal projections (i.e., connections that link different cortical areas). The pyramidal neuron was carefully sculpted to accurately reflect the true morphology observed in the brain. The key elements that carry specific neuronal functions such as dendrites, cell soma, and the axon were incorporated into the model. The sculpting process accounted for the different types of neurons found in various layers of the cortical column. Following this, materials and textures were applied to enhance the representation of the neuron and its surrounding context, as shown in Figure 7C. Subsequently, the animation was rendered and exported as high-quality frames, which would later be compiled using video editing software.

### 3.4. Final Video

The final video is rendered in After Effects and kept within 1 min and 30 s, to ensure that the duration is optimized to sustain the viewer’s attention. Clear annotations throughout the video guide the viewer to additional information as shown in Figure 6A. This immersive short video journeying through the brain engages students and viewers in a unique way by revealing the brain’s different organizational levels. Thus, the viewer can observe a macroscopic view of lobes and hemispheres to the microscopic intricacies of cortical columns and neurons. This visual exploration provides an engaging and exciting learning experience, which also builds a better understanding and appreciation for the brain’s complexity.

### 3.5. Classroom Study Results

To evaluate the effectiveness of the video on student comprehension and engagement, we conducted a randomized controlled trial. We asked whether watching the video visualization of the brain to supplement the classroom lecture material enhanced learning and engagement in students. Figure 8 shows box plots representing pretest and posttest scores of the survey questions in the control and experimental group. Learner data can be found in Appendix A, while the pre–post measures can be found in Appendix A. The median test score is represented by the horizontal line inside each box.

We were interested in revealing students’ perception of the delivery method (lecture or lecture and video), and their motivation to improve their neuroscience literacy after viewing the video. The Likert scale ratings in Figure 9 illustrate the reflections of students in the control and experimental groups, regarding their mode of delivery. In the posttest responses, 51% of students in the control group (Figure 9A, top panel) indicated agreement or strong agreement with the question “How motivated are you to deepen your understanding of the complex nature of the brain?” In contrast, two-thirds of students (66%) in the experimental group either agreed or strongly agreed with the same question (Figure 9B, top panel). Similarly, when students in the control group were asked in the posttest about their current understanding of the general organization of the brain, 24% of students responded with “great” or “excellent” (Figure 9A, bottom panel). However, after watching the video, 50% of students in the experimental group reported their current understanding of the general organization of the brain as “great” or “excellent” (Figure 9B, bottom panel).

Additionally, when students in the experimental group were asked in the posttest survey about their preferred medium for understanding complex topics, 61.9% of students reported that videos and interactive visualizations were their preferred mediums (Figure 9C). These results indicate that integrating videos with lectures could potentially boost students’ motivation to deepen their understanding of neuroscience and improve their perception of their own understanding of complex brain concepts. Furthermore, the preference for videos and interactive visualizations reflects the potential effectiveness of these mediums in facilitating students’ comprehension of complex concepts in neuroscience.

The GEE (generalized estimating equations) analysis (Table 1, Figure 10) indicated that participants in the experimental group had significantly greater odds of giving higher ratings compared to the control group (OR = 1.91, 95% CI [1.04, 3.51], *p* = 0.038). Across both groups, posttest ratings were significantly higher than pretest ratings (OR = 2.11, 95% CI [1.50, 2.96], *p* < 0.001). The group-by-time interaction was not statistically significant (OR = 1.07, 95% CI [0.65, 1.74], *p* = 0.801), indicating that the relative pre–post change did not differ significantly between groups when pooling across items. Likert items 2 and 3 had higher odds of elevated ratings compared to item 1 (OR = 1.58 and OR = 3.57, respectively, both *p* < 0.01), reflecting consistent differences in item-level response patterns.

The Wilcoxon signed-rank tests (Table 2) provided an item-level assessment of within-group changes. In the control group, Likert 1 increased significantly from a median of 2 to 3 (*p* = 0.0001), and Likert 2 showed a smaller but statistically significant increase from 3 to 3 (*p* = 0.0325). Likert 3 rose from 3 to 4, but this change was not significant (*p* = 0.227). In the experimental group, Likert 1 increased from a median of 3 to 4 (*p* < 0.0001), and Likert 2 showed a modest but significant improvement from 3 to 3 (*p* = 0.0465). Likert 3 remained unchanged at a median of 4 (*p* = 0.218).

Figure 11 shows the proportion of participants in each group providing high-agreement ratings (scores of 4–5) posttest for each Likert item. The experimental group consistently reported higher proportions than the control group, with the largest difference for item 1, where nearly half of experimental participants gave high ratings compared to about one-fifth in the control group. These patterns are consistent with the GEE and Wilcoxon results, supporting the conclusion that the intervention produced meaningful improvements in perceived outcomes.

Findings from both the population-level GEE model and the item-level Wilcoxon tests suggest that the intervention was associated with higher posttest ratings and greater positive change in the experimental group, particularly for the first two items. Although the GEE interaction term was not significant, descriptive patterns and within-group results consistently favored the experimental group.

## 4. Discussion

In the current work we introduced a novel pedagogical tool in the form of a video, taking the viewer on an immersive journey through the brain. This was achieved by using advanced 3D modeling and visualization tools. We documented the procedure and parameters so others can reproduce it. Lastly, we evaluated the effectiveness of the video by conducting a short brain knowledge survey in the classroom, in which students were randomized into a control and experimental group. The control group received a short lecture while the experimental group received a lecture plus watched the video. The video group showed larger pre–post gains and higher motivation than those receiving a lecture alone. This suggests that a short 3D video is an effective supplement to lectures in this setting.

Neuroscience explains how brain structure and function shape cognition and behavior [1,2,3,34]. Studies focusing on normal brain [35,36,37,38,39,40] are particularly valuable as they identify periods vulnerable to atypical sensory experiences. Abnormal visual experience can affect development, potentially leading to neurodevelopmental disorders [41,42]. This emphasis on neuroscience research underscores its tremendous societal value. These findings underscore the field’s societal value and the need for strong neuroscience education. Classroom practice benefits from brain-based insights that inform teaching and learning [11,12,13].

With rapid advancements in modeling and visualization software, there is an extraordinary opportunity to leverage their full potential. These tools, previously harnessed in areas like astrophysics [18] and biology [43], can really transform how we create neuroscience visuals. Leveraging software like Rhino 6, Houdini, and Cinema4D, we have bridged MRI scans with digital brain reconstructions, ensuring accuracy and visual appeal. Our video not only clarifies content with embedded annotations but also engages the viewer as the brain is rendered at the beginning of the video.

There are several advantages to using these 3D modeling and visualization tools in creating neuroscientific content. The software offers the flexibility of customizing important details in the production of videos and images, ultimately enabling the user to mimic the actual structure of the brain, circuits, and neurons. Moreover, the software’s annotation and labeling feature within its video-editing component is invaluable for students. Students can learn and understand concepts at their own pace, and thus are not restricted to learning in the traditional classroom environment. Lastly, the software used to produce the 3D models and video is heavily used in engineering and technical education domains. As the value of these tools becomes more evident, there is growing interest in the scientific community. These tools are recognized for their potential to create interactive and engaging educational content, making learning more immersive and effective.

However, there are several known limitations of using these advanced tools, which include specialized training and computational demands. For example, Rhino, Houdini, and Cinema4D require paid licenses, which can limit adoption and exact replication, while Fiji and MeshLab are open-source. Houdini’s node workflow has a steep learning curve, and Rhino requires prior CAD skills, which increases training time. Final renders need a modern GPU and time for sampling and denoising, so classrooms without this capacity should use pre-rendered video. Lastly, software updates can change filter behavior or default node settings, even in scripted workflows. Our results come from one course at one institution and do not cover other courses or XR delivery. We list software versions and key parameters to limit version drift and help others replicate the work.

In this study, we deployed the video in a 96-student course as a brief add-on to lecture. Our contribution is integration and transparency: we describe a fully specified, reproducible pipeline from MRI to mesh to animation to classroom test, disclose parameters, and include expert validation. We rely on peer reviewed tools for segmentation/registration and mesh processing [44], as well as validate gross anatomy and microanatomy using established atlases [1].

Multiple controlled studies and meta-analyses show VR or interactive 3D visualization outperforms traditional 2D resources for understanding spatial neuroanatomy, especially when used as a supplement [45,46,47,48,49]. For example, recent studies have reported higher posttest scores or spatial gains after VR lessons or 3D displays compared to conventional slides [50]. Our data are consistent with these findings in that the video and lecture group showed larger pre–post improvement and higher motivation than the group receiving a lecture alone. Furthermore, 3D and XR tools are also adopted clinically for neurosurgical planning and rehearsal. Hanalioglu and colleagues [51] recently reported improved anatomical visualization and planning accuracy using patient-specific 3D models and mixed/virtual reality systems in neurosurgical residents [51]. Our work provides a clear template to reproduce 3D instructional videos for classroom delivery with measurable outcomes, for adoption across courses and institutions.

## Figures and Tables

**Figure 1 jimaging-11-00298-f001:**
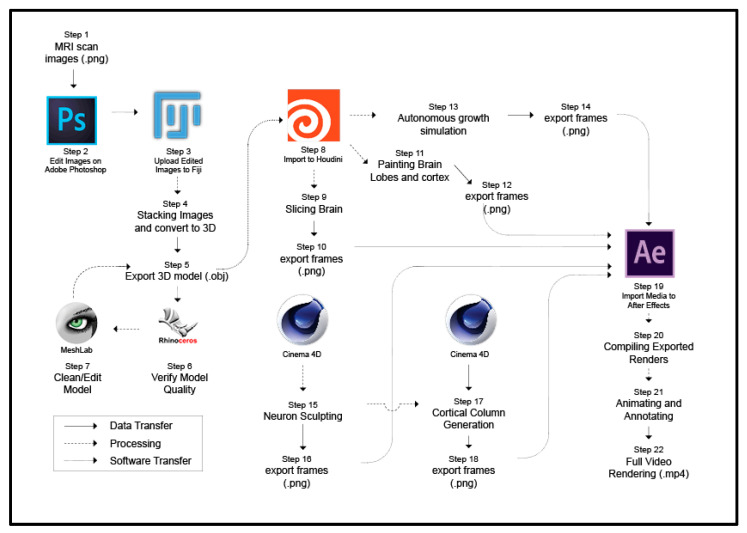
Workflow from source images to the final video. The diagram shows all twenty-two steps required for modeling, editing, animating, and rendering of the brain structures presented in the video. Arrows indicate data flow.

**Figure 2 jimaging-11-00298-f002:**
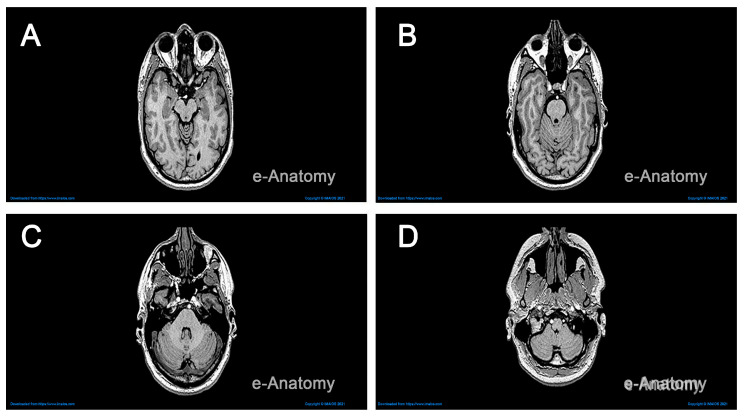
MRI scans in the horizontal plane revealing major brain structures. Panels (**A**–**D**) correspond to slice indices 80, 91, 102, and 121. Each slice displays cerebral hemispheres, ventricles, brainstem segments, and cerebellum as visible at that level. Window and level settings remain constant across panels for comparison (Step 1).

**Figure 3 jimaging-11-00298-f003:**
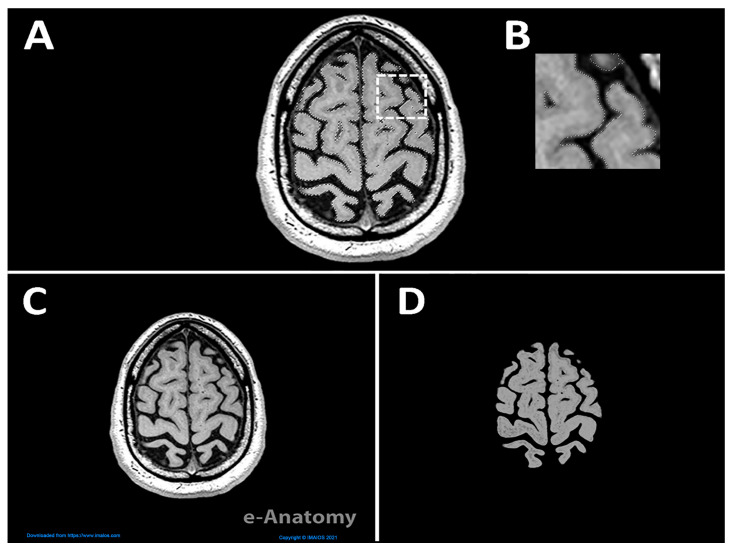
Adobe Photoshop workflow for selection and cleanup. (**A**) Image in Adobe Photoshop showing the highlighted section of the brain. The dashed white square marks the area enlarged in (**B**). (**B**) Higher magnification of the boxed regions showing the selected boundaries after refinement. (**C**) Raw image downloaded from IMAIOS, and (**D**) edited image after selection and cleanup, with background and clearer edges compared to C used to create the 3D object of the brain (Step 2).

**Figure 4 jimaging-11-00298-f004:**
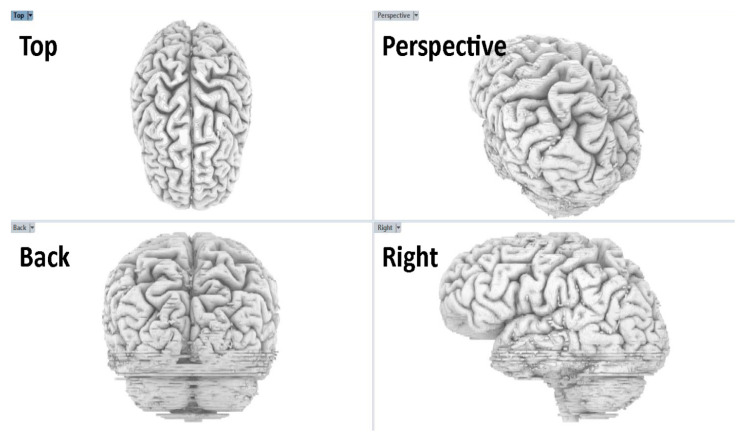
Rhino 6 four-viewport review of the brain model. Top, perspective, back, and right views show global shape and sulcal–gyral patterns. We confirmed scale and orientation as well as checked for symmetry. This review step preceded animation and export (Step 6).

**Figure 5 jimaging-11-00298-f005:**
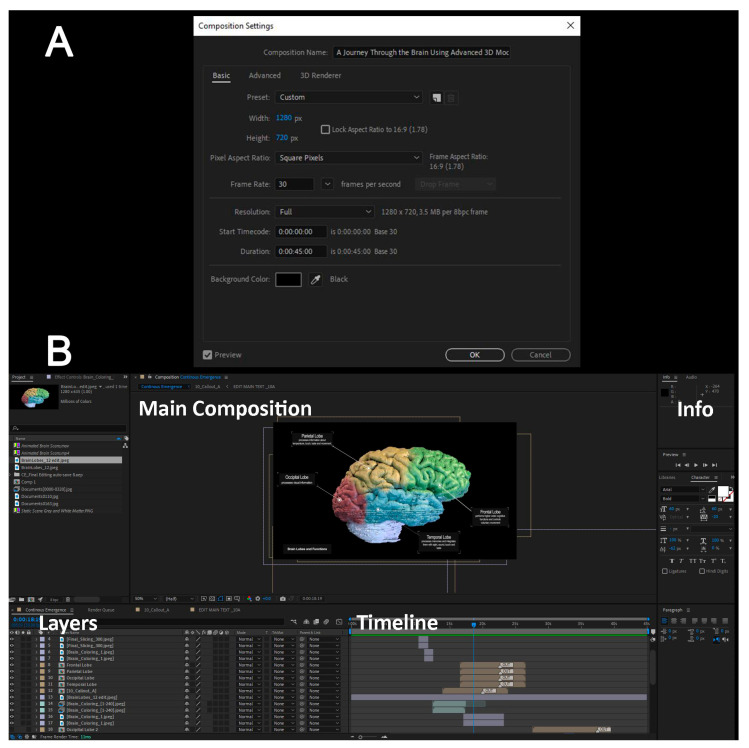
Post-production setup for the final video in After Effects (Step 17). (**A**) Composition settings including frame size, frame rate, and color management. (**B**) Shows the layer stack used for titles, labels, overlays, and color adjustments (Step 19).

**Figure 6 jimaging-11-00298-f006:**
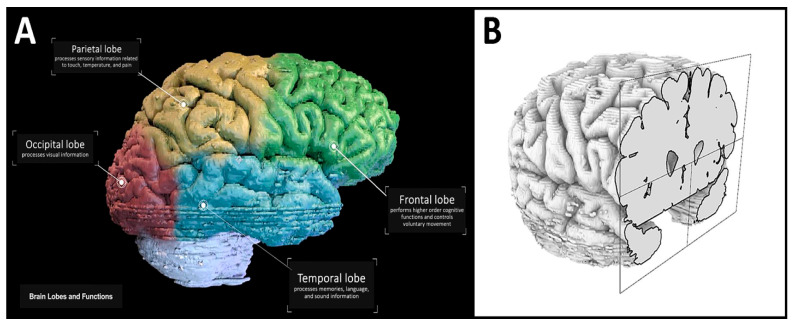
Brain lobes and a coronal section view. (**A**) Labeled lobes and their corresponding functions in the final video render (Step 19). (**B**) Visualization of the imported brain model with a coronal cut to reveal internal anatomy (Step 6).

**Figure 7 jimaging-11-00298-f007:**
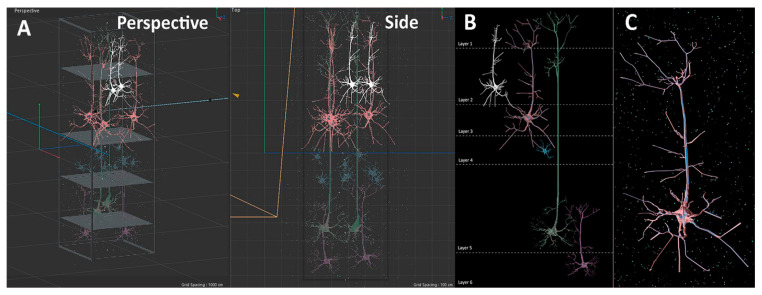
Cortical layers and a sample pyramidal neuron. (**A**) The different layers and accompanying cells modeled in a cortical column. (**B**) Full cortical column visual in Cinema4D. (**C**) A representative pyramidal neuron from layer 3 after the sculpting process in Cinema4D. The neuron includes soma, basal and apical dendrites, and axon (Step 15).

**Figure 8 jimaging-11-00298-f008:**
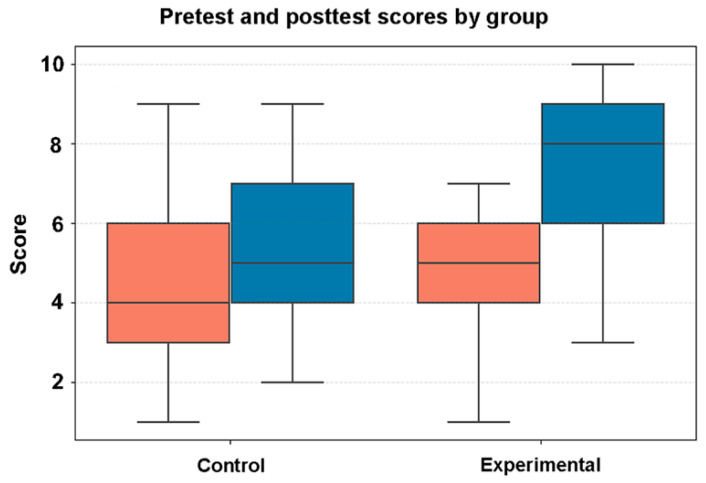
Student test scores on the brain knowledge assessment. Boxplots show the distribution per study arm. Orange boxes show pretest results; blue boxes show posttest results. Left pair = control (*n* = 46). Right pair = experimental (*n* = 50). Boxes represent the interquartile range, the line is the median value, and whiskers extend to 1.5 × IQR. Scores are on a 0–10 scale. Statistical comparisons are reported in the Results.

**Figure 9 jimaging-11-00298-f009:**
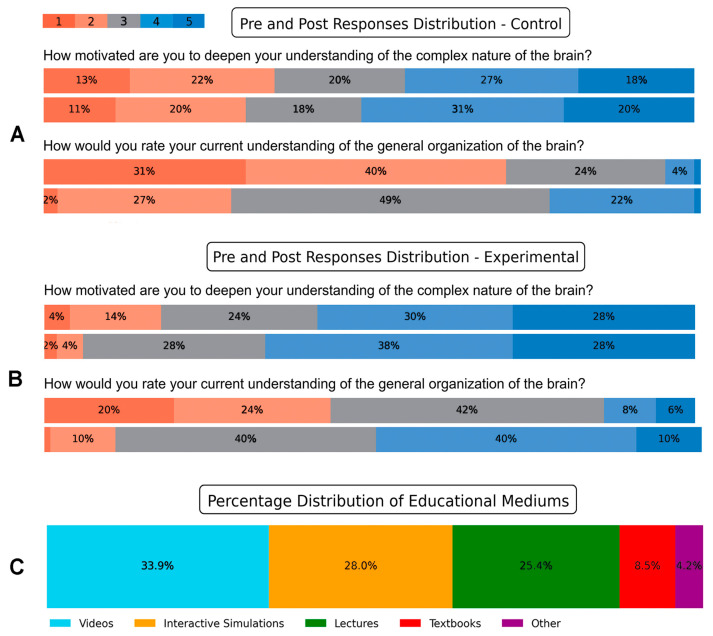
Distributions of Likert responses and preferred learning media. Responses of representative Likert scale questions in the (**A**) control and (**B**) experimental group. For each question, the top panels are pretest responses, and the bottom panels are the posttest responses. Colors map scores 1–5 left to right. Segment labels show percentages. Higher scores indicate stronger agreement. (**C**) Posttest responses of preferred educational mediums by students in the experimental group.

**Figure 10 jimaging-11-00298-f010:**
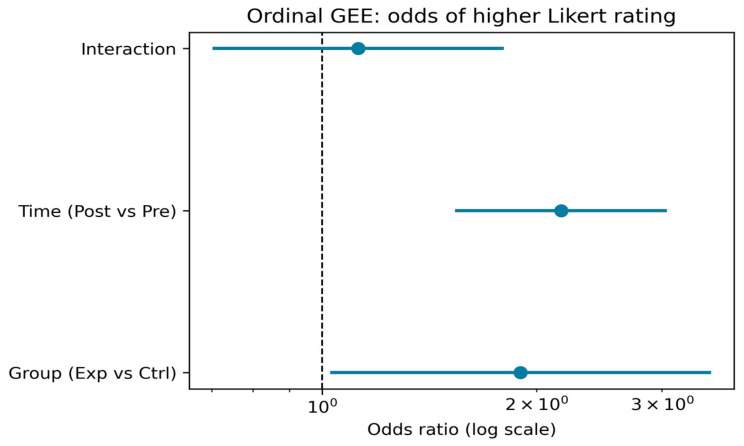
Odds ratios and 95% confidence intervals from the ordinal generalized estimating equations (GEE) model, estimating the likelihood of higher Likert ratings. The model includes main effects for group (experimental vs. control) and time (post–vs. pre–test), as well as their interaction. An odds ratio above 1 indicates increased odds of a higher rating for the given comparison.

**Figure 11 jimaging-11-00298-f011:**
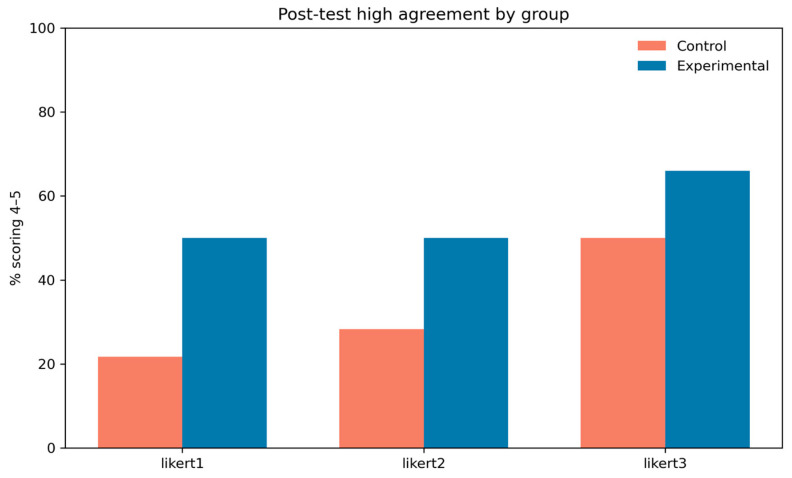
Percentage of participants scoring in the high-agreement range (Likert ratings 4–5) post test for each item, stratified by group (control vs. experimental). This provides a descriptive comparison of post-intervention agreement levels across the three Likert items.

**Table 1 jimaging-11-00298-t001:** Proportional-odds model of Likert ratings: odds ratios (95% CI) and *p*-value.

Term	Odds Ratio	95% CI (Lower)	95% CI (Upper)	*p*-Value
Group (Exp vs. Ctrl)	1.91	1.04	3.51	0.038
Time (Post vs. Pre)	2.11	1.5	2.96	<0.001
Interaction	1.07	0.65	1.74	0.801
Likert2 vs. Likert1	1.58	1.19	2.09	0.001
Likert3 vs. Likert1	3.57	2.42	5.28	<0.001

**Table 2 jimaging-11-00298-t002:** Within-group median changes in Likert items pre versus post with Wilcoxon *p*-values.

Group	Item	Median Pre	Median Post	*p*-Value
Control	Likert 1	2	3	0.0001
Control	Likert 2	3	3	0.0325
Control	Likert 3	3	4	0.227
Experimental	Likert 1	3	4	<0.0001
Experimental	Likert 2	3	3	0.0465
Experimental	Likert 3	4	4	0.218

## Data Availability

The video is available at https://figshare.com/s/5fd9c530ada5515b7d35 and all the original files are publicly available on Zenodo (https://zenodo.org/records/16892279). Accessed on 25 July 2025.

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
