# Peer review of "From Brain Lobes to Neurons: Navigating the Brain Using Advanced 3D Modeling and Visualization Tools"

_2313-433X, 2025, doi:10.3390/jimaging11090298_

Round 1

Reviewer 1 Report

Comments and Suggestions for Authors

Summary: This paper presents a novel educational tool combining neuroscience with 3D modeling and visualization to enhance learning. Using MRI data and software like Fiji, Rhino, Houdini, and Cinema4D, the authors created an animated video illustrating brain structures from lobes to neurons. A classroom experiment with 96 students showed improved comprehension and motivation in the group that watched the video. The study highlights the potential of visual tools in science education and offers a reproducible workflow. While well-executed, minor improvements in methodological transparency and model validation would further strengthen its impact and applicability.

Recommendation with minor revision:

  • While the paper implies the hypothesis that 3D visualization improves learning, it would benefit from a clearly stated hypothesis or research question in the Introduction.
  • 2nd page is empty, focus on journal’s template.
  • The methodology lacks a formal validation step comparing the generated 3D brain model to a ground truth or benchmark (e.g., anatomical atlas). Write theoretical explanation, such as mathematical structure.
  • The description of parameter settings (e.g., smoothing thresholds in MeshLab, camera angles in Houdini, lighting intensity) is limited. Including a supplementary file or table with these details would significantly improve reproducibility.

Reviewer 2 Report

Comments and Suggestions for Authors

The paper developed an educational video using 3D modeling and visualization software, including Rhino 6, Houdini FX, and Cinema4D. The video, created using MRI data and other anatomical references, provides an immersive exploration of the brain’s macro- and microstructures. Te authors relied on student feedback  to show  a significant improvement in comprehension and motivation among those who viewed the video compared to traditional lecture formats.

This video provides 3D modeling and visualization tool to empower science education by enhancing student engagement and improving the understanding of complex scientific concepts. However, there are few issues required to be clarified as follows:

1-The authors should explain the limitation and challenges of the application tool compared to other existing tools.

2- It is recommeded to add background on visualization software, including Rhino 6, Houdini FX, and Cinema4D.

3-Figure 1. Workflow of the modeling process. For improved clarity, the implementation steps should be explicitly numbered within the figure (e.g., Step 1, Step 2, Step 3, etc.) to clearly illustrate the sequential process. Additionally, it is recommended to present these steps in the main text as a bulleted list to enhance readability and understanding.

4- In Figure 5, the slide responses to the questions presented in panels (A) and (B) do not clearly reflect the students' feedback in the Control and Experiment modes. Please provide clarification regarding the percentages shown, as well as the meaning of the colors used in these slides (1,2,3,4,5).

5-The video (https://figshare.com/s/5fd9c530ada5515b7d35) which is mentoned in Section 3.1, should be added at the end of manuscript as online supplemenrty video.

Reviewer 3 Report

Comments and Suggestions for Authors

Round 2

Reviewer 2 Report

Comments and Suggestions for Authors

Thank you for the opportunity to review this revision. I complement the authors for their thorough and extensive revision. All the specific and general review comments have been properly addressed.

I would only suggest changing the text color of the numerical percentage values in Figure 9 from white to black to improve readability. (e.g. 10%, 28%, etc.)

Author Response

We thank the reviewer once more for their effort and time in reviewing the manuscript. We have modified figure 9 according to the reviewer's suggestion

Reviewer 3 Report

Comments and Suggestions for Authors

The authors have taken my recommendations into account, I have no other comments.

Author Response

We thank the reviewer for their time and effort in reviewing the manuscript.